# Acclimation and Compensating Metabolite Responses to UV-B Radiation in Natural and Transgenic *Populus* spp. Defective in Lignin Biosynthesis

**DOI:** 10.3390/metabo12080767

**Published:** 2022-08-20

**Authors:** Tiffany M. Wong, Joe H. Sullivan, Edward Eisenstein

**Affiliations:** 1Institute for Bioscience and Biotechnology Research, University of Maryland, 9600 Gudelsky Drive, Rockville, MD 20850, USA; 2Department of Plant Science and Landscape Architecture, University of Maryland, College Park, MD 20742, USA; 3Fischell Department of Bioengineering, University of Maryland, 9600 Gudelsky Drive, Rockville, MD 20850, USA

**Keywords:** UV-B radiation, *Populus*, acclimation, lignin, secondary metabolites, targeted metabolomics, phenylpropanoids, anthocyanins, salicylates

## Abstract

Plants have evolved to protect leaf mesophyll tissue from damage caused by UV-B radiation by producing an array of UV-absorbing secondary metabolites. Flavonoids (phenolic glycosides) and sinapate esters (hydroxycinnamates) have been implicated as UV-B protective compounds because of the accumulation in the leaf epidermis and the strong absorption in the wavelengths corresponding to UV. Environmental adaptations by plants also generate a suite of responses for protection against damage caused by UV-B radiation, with plants from high elevations or low latitudes generally displaying greater adaptation or tolerance to UV-B radiation. In an effort to explore the relationships between plant lignin levels and composition, the origin of growth elevation, and the hierarchical synthesis of UV-screening compounds, a collection of natural variants as well as transgenic *Populus* spp. were examined for sensitivity or acclimation to UV-B radiation under greenhouse and laboratory conditions. Noninvasive, ecophysiological measurements using epidermal transmittance and chlorophyll fluorescence as well as metabolite measurements using UPLC-MS generally revealed that the synthesis of anthocyanins, flavonoids, and lignin precursors are increased in *Populus* upon moderate to high UV-B treatment. However, poplar plants with genetic modifications that affect lignin biosynthesis, or natural variants with altered lignin levels and compositions, displayed complex changes in phenylpropanoid metabolites. A balance between elevated metabolic precursors to protective phenylpropanoids and increased biosynthesis of these anthocyanins, flavonoids, and lignin is proposed to play a role in the acclimation of *Populus* to UV-B radiation and may provide a useful tool in engineering plants as improved bioenergy feedstocks.

## 1. Introduction

Research on woody plants offers promise for the development of next-generation biofuel feedstocks with reduced lignin recalcitrance and enhanced saccharification for ethanol production. It has long been recognized that a key barrier to the profitable use of plant feedstocks for biofuel is the removal of lignin from cellulose, enabling cellulose saccharification processes to produce monosaccharides for effective and efficient conversion to ethanol. Thus, early work in the removal of lignin for paper products to improve cellulose pulp eventually led to research into the first-generation biofuel feedstocks [1].

Lignin is a key, indispensable polymer in plants. It provides strength for plants to grow tall in gravitropic environments and provides a hydrophobic lining to facilitate water and nutrient transport in plant vasculature. Lignin is the second most abundant polymer on Earth, comprising up to 30% of organic carbon in the biosphere. Lignin assembles as a large network of aromatic precursors that polymerize via oxidative coupling of monolignols (hydroxycinnamyl alcohols, including coniferyl alcohol, sinapyl alcohol, and *p*-coumaryl alcohol) via a radical mechanism. Monolignols are products of the phenylpropanoid biosynthetic pathway, which also produces the primary UV-absorbing secondary metabolites in plants, including phenols and flavonoids [2,3,4,5]. Efforts to engineer lignin levels or composition have shed light on these complexities and revealed potential unanticipated outcomes [6,7]. On the other hand, the relationships between plant lignin levels and composition, the origin of growth elevation, and the hierarchical synthesis of UV-screening compounds are additional, important considerations to engineering lignin levels, but these connections remain unclear. Interestingly, recent studies have revealed that lignin found in the plant cell walls is not only integral for structural support but provides protection against UV-B radiation [8,9]. The impact of quantitative and qualitative differences between these components and how it relates to UV tolerance or sensitivity in plants is unknown.

*Populus* trees grow at a wide range of elevations with tremendous variation in natural lignin content and composition, and thus, *Populus* spp. may provide a useful model for biofuel research and breeding programs to improve cellulosic degradability. Secondary cell walls in plants are important for their structural integrity as they comprise a dense matrix of polymers that provide strength and rigidity. Because lignin comprises a primary component of these polymers, the downregulation of lignin precursor genes in the phenylpropanoid biosynthetic pathway in transgenic *Populus trichocarpa* has been explored in order to reduce lignin content and improve cellulosic biomass degradability for biofuel production, with encouraging results [10,11]. Interestingly, some natural poplar variants can also release unusually high sugar yields without undergoing chemical pretreatment during cellulosic biomass conversion to fuel [12]. Consequently, *Populus deltoides* has been modified to enhance saccharification not only by modifying genes in the phenylpropanoid pathway but also genes involved in cell wall development, resulting in plants’ altered lignin levels and improved cellulose release [13].

The monolignols coumaryl alcohol, coniferyl alcohol, and sinapyl alcohol are derived from the phenylpropanoid pathway and serve as the source of secondary metabolites such as anthocyanins, tannins, and other aromatic compounds that function in plant signaling, defense, and abiotic stress responses [14,15,16]. However, the reduction of lignin in a variety of plants, including *Populus* with suppressed enzymes important in the phenylpropanoid pathway, demonstrated a range of abnormal growth phenotypes with reduced fitness and may display increased sensitivity to UV-B [17,18,19]. It is possible that *Populus* variants may compensate for low lignin levels when acclimating to UV-B radiation exposure by increasing the production of other aromatic specialized compounds to alleviate the impact and stress of UV-B [2,3,4]. Therefore, not only is the phenylpropanoid biosynthetic pathway important for lignin biosynthesis, but it is also of central importance to *Populus* physiology, plant defense, and overall health and fitness. Here we examine representative *Populus* collections for correlations between sensitivity (or acclimation) to UV-B radiation under greenhouse and laboratory conditions, or poplars with different levels of lignin, whether arising from variation in natural populations that thrive under different conditions or from intentional modification of genes in transgenic lines. Natural *Populus* variants from the low and high origin of growth elevation that possess altered lignin content suggest that UV protection is decreased in plants with lower lignin levels, consistent with work describing the absorption of UV by lignin [14,19,20,21]. Additionally, transgenic *Populus* variants with modifications in phenylpropanoid and lignin biosynthesis were used to examine the hypothesis that precursors and lignin levels correlate with UV-B protection. Noninvasive, ecophysiological, and metabolite measurements revealed that the synthesis of anthocyanins, flavonoids, and lignin precursors is increased in *Populus* upon exposure to UV-B radiation. Furthermore, plants with genetic modifications that affect lignin biosynthesis, or natural variants with altered lignin levels and compositions, display complex changes in phenylpropanoid metabolites, but generally, the biosynthesis of these protective compounds is increased stemming from *Populus* acclimation to UV-B radiation.

## 2. Experimental Design and Methods

***Plant Material***—A natural population of 25 *P. trichocarpa* (Torr and Gray) genotypes were obtained from the Department of Biology at West Virginia University and the BioEnergy Science Center (currently, Center for Bioenergy Innovation) and are summarized in Table 1. *Populus trichocarpa* trees were previously sampled along the Pacific Northwest and geographical information was recorded [22]. It is important to note that although lignin plays a role in vascular and structural support, it is predominately concentrated in the shoots, with some lignin in leaf tissue [23]. Leaves were marked at the beginning of treatment prior to UV experimentation and monitored throughout the duration of treatments. The same leaves were then harvested and flash frozen in liquid nitrogen after UV experimentation for further analysis. *Populus* were maintained and grown under fluorescent lamps for 16 h at room temperature. The various genotypes ranged from low to high syringyl (S) to guaiacyl (G) lignin ratios (S/G) as well as from low to high lignin content (%). Lignin compositions were obtained from literature results of biomass analysis of wood cores and quantification of sugar release from a natural population of *P. trichocarpa* [12]. The range of lignin content varied from 15.7% (low) to 27.9% (high), and the S/G compositional was defined in this study as low at <2.0 and high at >2.0. The plants used in this study have been previously characterized by Muchero et al. [22] and were collected from the same region, with collaborators in the same laboratory at the BioEnergy Science Center (BESC) obtaining the values describing lignin chemistry [22].

Additionally, as summarized in Table 2, a collection of transgenic *P. deltoides* modified in cell wall development was also obtained and propagated in the greenhouse under similar conditions as previously described [24]. Transgenic *P. deltoides* lines were obtained from BESC that targeted genes that increased cellulose or reduced crystallinity, altered hemicellulose composition, or modified enzymes in the phenylpropanoid pathway in an effort to reduce recalcitrance during degradation of lignocellulosic biomass [13]. Transformation of *P. deltoides* (Marsh.) clone, WV94 from Issaquena County., MS. was performed by ArborGen LLC. as described previously [25] and obtained from the Department of Biology at West Virginia University.

All *Populus* cuttings were propagated and planted in 6.0 L pots filled with soilless growing media (Sunshine LC1) and maintained in the greenhouse at the University of Maryland (College Park, MD, USA) at 25 °C with 16 h photoperiod. Photosynthetic photon flux (PPF) between 400–700 nm was approximately 1000 μmol m^−2^ s^−1^ at midday. Temperatures were controlled by an evaporative cooling system and heating element to maintain 25 °C. Representative plants (*n* = 3) similar in height and appearance were based on availability and used for experimentation for each genotype and treatment condition.

***UV-B Treatments***—Lamp banks of UV-B fluorescent tubes were suspended above greenhouse benches. The lamp bank contained twelve Q-lab UVB-313 QFS40W fluorescent sunlamps (Q-panel Inc., Westlake, OH, USA). Lamps were wrapped with 0.038 mm cellulose acetate film to absorb radiation below 290 nm (UV-B treatment) or 0.051 mm clear polyester (Mylar) film to absorb radiation below 315 nm (control treatment) following the general procedure outlined in Sullivan and Teramura [24]. Each fluorescent UV-B tube was measured with a broadband Solarmeter Digital Ultraviolet Meter 6.2 UV-B meter (Solartech, Harrison Township, MI) in μW cm^−2^, and recorded values were converted to kJ m^−2^ at 30 or 60 cm. UV-B experiments involved UV-B dosage ranging from 10–35 kJ m^−2^ for 6–8 h centered around solar noon for 10–14 days.

***Epidermal UV Transmittance***—Measurement of the degree of epidermal UV transmittance via fluorescence in leaf tissue is an indirect method that relates metabolic changes in epidermal UV-absorbing compounds to UV exposure. UV-absorbing metabolites produced in plants can reduce radiation penetration into mesophyll tissue with resulting changes in UV-promoted fluorescence of chlorophyll measured with a UV-A PAM instrument. Thus, UV-promoted changes in epidermal chemistry can be correlated with alterations in chlorophyll fluorescence. UV-screening and inferred epidermal shielding were measured with a portable field pulse amplitude modulation (PAM) chlorophyll fluorometer (UV-A PAM; Gademann Instruments, Wurzburg, Germany). The instrument measures the UV transmittance in the epidermis of leaves by comparing the ratio of chlorophyll fluorescence from UV-A (372 nm) divided by the chlorophyll fluorescence from blue light (470 nm) to calculate the relative extent of UV transmittance into the mesophyll. As illustrated in the schematic presented in Figure 1A, the UV-A PAM approach provides a noninvasive, sensitive, and accurate measure of UV transmittance by measuring chlorophyll fluorescence that is proportional to UV shielding. Transmission results can be used to correlate chlorophyll fluorescence to epidermal shielding compounds in leaf tissue [26]. However, the UV-A PAM approach only infers concentration, and in addition, screening compounds in the mesophyll may not be detected, requiring further analysis by LC-MS to provide a quantitative analysis of UV-screening compounds. The instrument uses epidermal UV-A transmittance to excite chlorophyll, with the resulting fluorescence proportional to the incident radiation. UV screening is then estimated by the ratio of fluorescence excited at 375 nm and 470 nm [27]. Chlorophyll fluorescence is normalized by exciting chlorophyll molecules with blue light at 470 nm that is not absorbed by UV-absorbing compounds in the epidermis [28,29]. The UV-A PAM has been compared to UV transmittance in epidermal peels and found to be correlated with intact leaf samples, thereby allowing for data collection nondestructively through experimentation [30]. A decrease in UV transmittance corresponds to an increase in UV protecting compounds that accumulate in the epidermis, and the percentage of chlorophyll fluorescence is decreased.

The leaf plastochron index (LPI) range was used to assign an arbitrary age based on a morphological time scale rather than chronological time [31]. Three moderately expanded leaves were selected at the top canopy, marked, and measured with the liquid light guide at the middle section from the midrib for each pot. Ten repeated measurements were made on each leaf and averaged to calculate a single response on each plant each day. Adaxial surfaces of leaves at LPI 2–3, LPI 4–6, and LPI 7–8, corresponding to a range of younger to older leaves, respectively, were analyzed. These leaves were marked and selected for UV transmittance measurements with the pulse amplitude modulation (PAM) chlorophyll fluorometer described above and for chlorophyll fluorescence measurements (below). Typically, UV-A PAM measurements were taken every other day with ten repeated measurements made on the same three leaves throughout experimentation and averaged with standard error calculation.

***Chlorophyll Fluorescence Measurements***—UV-B damage to photosynthetic machinery was measured by chlorophyll fluorescence [32]. Measuring chlorophyll fluorescence offers a non-destructive approach to estimating plant stress by estimating the plant’s ability to utilize light to oxidize water molecules by converting it into chemical energy in the photosynthetic reaction center [33]. UV-B radiation has been shown to affect photosynthetic efficiency [34], and this can be estimated in plants by evaluating the sensitivity to UV-B radiation and then represented as a ratio of available open and closed reaction centers in the thylakoid membrane, as shown schematically in Figure 1B. A low photosynthetic efficiency ratio arises when photosystem II (PSII) contains fewer open reaction centers due to damage and impairment of the Calvin cycle [35]. The measurement relies on estimating the energy flow through reaction centers after 20–30 min of dark acclimation [36]. A portable chlorophyll fluorometer (OS30p+, Opti-Sciences Inc., Hudson, NH, USA) was used to measure the minimal fluorescence (Fo) and the maximum fluorescence (Fm) of chlorophyll that is used to calculate the potential maximum photosynthetic efficiency as the ratio of variable fluorescence (Fv) to maximum fluorescence as Fv/Fm [36]. Two measurements were made on dark-adapted leaves at various LPI on opposite sides, avoiding the midrib prior to experimentation and immediately following the last day of UV-B treatment. An iterative approach was used to establish optimal conditions for detecting metabolic changes in these *Populus* collections when treated with UV-B radiation. Initially, a noninvasive approach was used to measure the epidermal transmittance of UV in order to preserve plant specimens. Then, changes in the levels of any protective compounds were then further analyzed with LC-MS (see below). The maximum potential quantum efficiency of PSII was measured by the normalized ratio of variable fluorescence divided by maximum fluorescence (Fv/Fm). The measurement determines if photosynthetic efficiency is affected by plant stress, such as UV-B radiation. Previously selected leaves from the UV epidermal transmittance measurements were dark adapted for 20 min, and multiple locations were measured with a chlorophyll fluorometer. The instrument uses red light emitting diodes between 700–750 nm that detect low noise fluorescence in minimal fluorescence (Fo) and saturating light levels for s maximum fluorescence (Fm).

**Figure 1 metabolites-12-00767-f001:**
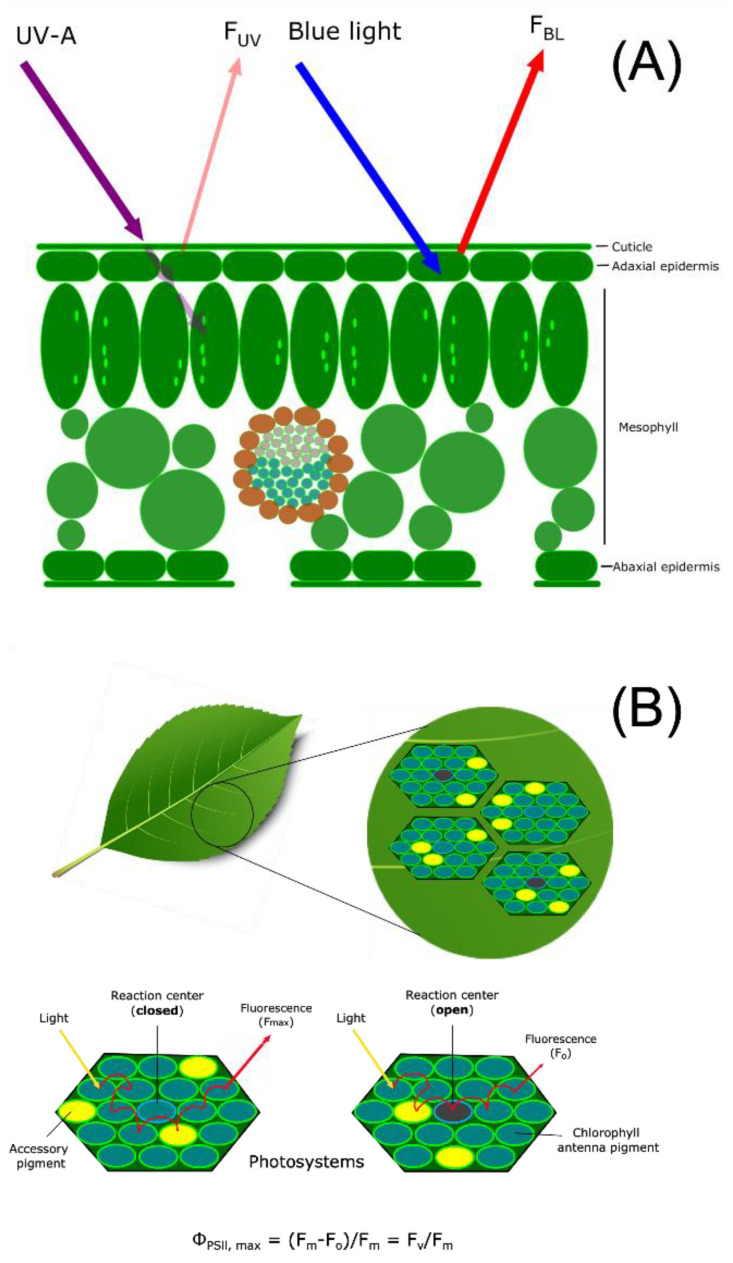
(**A**) Illustration of relative epidermal UV transmittance of chlorophyll fluorescence from UV-A radiation and blue light with the UV-A PAM chlorophyll fluorometer based on Bilger [37]. Excitation of chlorophyll from light-emitting diodes at 375 nm (UV-A) and 470 nm (blue light) is supplied by a 5 mm flexible tube with a liquid core light guide and a leaf clip attachment for a non-destructive measurement. A ratio is calculated by measuring chlorophyll fluorescence induced by UV (F_UVA_) and visible blue light reference (F_BL_) on the adaxial epidermal layer. Influence of UV-absorbing compounds accumulating in the epidermis will decrease F_UVA_ as less UV is absorbed by the leaf and re-emitted to the fluorometer. To determine the percent of UV transmittance or passing through the epidermis is calculated as UV-shielding (%) = 100 (1 − F_UVA_/F_BL_). (**B**) Schematic of the maximum quantum efficiency of Photosystem II (PSII) (Φ_PSII,max_) in open reaction centers measured with a chlorophyll fluorescence meter (F_v_/F_m_). Non-destructive fluorescence measurements are performed by dark-adapting plant samples followed by exposure to the saturating excitation light source from a light emitting diode. The maximum fluorescence (F_m_) represents available and oxidation state PSII reaction centers or when the level of fluorescence when the primary quinone electron acceptor has been maximally reduced and PSII reaction centers are closed. Leaf samples are exposed to saturating light intensity that closes and chemically reduces PSII reaction centers resulting in the maximum fluorescence value. The minimum fluorescence (F_o_) is measured by the level of fluorescence when plastoquinone is maximally oxidized and PSII centers are open resulting in reduced fluorescence emission [38]. F_v_/F_m_ can be used as a parameter of plant stress.

***Targeted Metabolomic Analysis***—Metabolite detection was performed on leaves previously selected for analysis using a UPLC-MS (Acquity Ultra Performance Liquid Chromatography system coupled with an LCT Premier XE Time-of-Flight Mass Spectrometer) system (Waters Corporation, Milford, MA, USA) [39]. Leaf tissue was flash frozen in liquid nitrogen, ground into a fine powder with a mortar and pestle, and stored at −80 °C until analysis. Secondary metabolites were extracted from 4 mg/leaf of ground tissue with 1.5 mL with HPLC-grade MeOH at room temperature for 30 min, centrifuged for 10 min at 2100× *g*, and filtered through microcentrifuge filtration units. Sample extracts were introduced into 12 × 32 mm screw neck glass vials with bonded pre-slit PTFE/silicone septa for automatic injection into a Waters Acquity C18 BEH 1.7 μm, 2.1 × 50 mm UPLC column (Waters Corporation, Milford, MA, USA) at 30 °C. The column mobile phase consisted of solvent A (water + 0.1% formic acid) and solvent B (acetonitrile + 0.1% formic acid) at a flow rate of 0.4 mL/min. Time of flight (TOF) mass spectrometry was set to negative electrospray mode with V resolution. Capillary and cone voltage was set to 2800 V and 80 V, respectively, and the scan range was between 150 to 800 *m*/*z*. All samples were analyzed with an internal reference lock mass using leucine enkephalin (200 ng/mL) at a flow rate of 2.0 μL/min. Chromatography and mass spectra were processed with the MassLynx 4.1 software package (Waters Corporation, Milford, MA, USA). Identification of signature compounds was achieved by comparison with internal and external standards (Millipore Corporation, St. Louis, MO, USA). Identification of compounds for which standards are unavailable was determined with LC-MS molecular weights and retention times from previous work [40,41] and the database within MassLynx. Using the mass of the deprotonated [MH] ion, single ion chromatograms were integrated to calculate peak areas and normalized to an internal standard, vanillic acid. Quercetin glucoside and salicin external standards were used to generate a standard curve to determine the concentration of each metabolite within a sample.

## 3. Results

In an effort to understand the impact of reduced lignin levels on UV-B acclimation in a biofuel feedstock, several specimens from the *Populus* collections listed in Table 1 and Table 2 were selected to represent limits of origins of growth elevation, lignin content and S/G ratio, and were examined for correlations between sensitivity (or acclimation) to UV-B radiation under greenhouse and laboratory conditions. Initial surveys revealed similar responses of variants with analogous characteristics, so the additional analyses described here were focused on only several specific natural and transgenic lines, as noted. Noninvasive, ecophysiological measurements of chlorophyll fluorescence in UV-B treated leaves were performed, and concentrations of key metabolites were determined by LC-MS.

***UV Sensitivity of Poplar Leaves at Different LPI***—Initially, optimal conditions were sought to determine the impact of UV-B radiation in *Populus* based on plant leaf age (measured by leaf plastochron index—LPI), time of exposure, and radiation intensity. Data were compared from a series of experiments using both *P. trichocarpa* and *P. deltoides* interchangeably since their responses to UV-B from the UVA-PAM, and Fv/Fm approaches were found to be similar (data not shown). Leaves at LPI 2–3, LPI 4–6, and LPI 7–8 of low-lignin *P. trichocarpa* were treated with UV-B radiation for seven days, and epidermal shielding was measured during the initial (I) and final (F) day of experimentation. Generally, a slight trend in an increase in epidermal shielding (decrease in epidermal transmittance of UV-B) was seen for both *P. trichocarpa* and *P. deltoides* that at low (~10 kJ m^−2^) levels of UV-B treatment, though the differences were qualitative and lacked statistical power to draw firm conclusions on the influence of leaf age on acclimation of low lignin *Populus* to UV-B radiation. However, at higher levels of radiation (~35 kJ m^−2^), epidermal screening was increased in both *P. trichocarpa* and *P. deltoides* by UV-B by the fifth day of treatment, with a maximal response observed after about 10 days of UV-B radiation as shown in Figure 2. Similarly, the maximum quantum efficiency of PSII in the dark-adapted state showed an increase in Fv/Fm in *P. deltoides* treated with UV-B compared to the control (data not shown). These preliminary experiments indicated that *Populus* leaves at LPI 4–6 could be marked and then analyzed on an average of 10 days after exposure to UV-B radiation at ~35 kJ m^−2^.

***Chlorophyll Fluorescence Response to UV-B in Populus Variants***—We sought to obtain some insight into any correlation between lignin content or composition, growth of elevation, and UV-B acclimation in the *Populus* variants under study, and therefore select plants in the collection were analyzed for 14 days, with UV transmittance recorded every other day to calculate relative change (%) from the first measurement. As can be seen in Figure 3A for variants obtained at low growth elevation, the relative chlorophyll fluorescence changes measured in a high-lignin variant (GW-10993, 27.20%, filled squares) showed a decrease in UV shielding relative to that seen in a low-lignin variant (BESC-99, 20.02%, open squares). This observation is consistent with the hypothesis that lignin, by absorbing UV light, helps protect the plants from UV-B radiation and therefore is an important component of the rate and/or extent of acclimation. The low elevation, high lignin plants displayed a reduced response to UV shielding of radiation relative to the low lignin plants, indicating that low-lignin plants are less well-adapted to UV-B radiation and, accordingly, mount a more robust response to UV-B stress. However, the similar analyses using *P. trichocarpa* variants obtained from higher growth elevations that were treated under the same conditions were different. As can be seen in Figure 3B, the relative change in chlorophyll fluorescence in a high-lignin variant (GW-9889, 27.10% filled triangles) showed an increase relative to that seen for a low-lignin variant (GW-9791; 20.38%, open triangles). In the case of high elevation poplars, higher levels of lignin appear less well adapted to UV-B radiation and therefore generate a greater response to radiation, whereas plants with low-lignin levels seem better adapted with therefore elicit only a more modest response.

Generally, UV-B radiation levels are typically higher at high elevation relative to low elevation; however, latitude, weather conditions, and ozone concentration can contribute to the variation of radiation intensity. Thus, to further probe the contrasting shielding results of poplars obtained from high and low elevation, data for the growth elevation of each *P. trichocarpa* genotype were compared to respective lignin chemistry data. These data were obtained for the natural variants of *P. trichocarpa* in the collection of plants previously characterized with respect to lignin chemistry from wood cores and geographical information [12]. The data for lignin composition and content were analyzed using regression analysis, and within a 65% confidence interval, neither parameter showed any dependence on growth elevation. These analyses are summarized in Figure 4A,B, and suggest that it may be hazardous to draw simple correlations between UV shielding and lignin levels in poplar plants from low and high elevations. Although lignin can provide protection and absorb UV-B radiation, there may be other unaccounted-for factors that provide protection. In addition, the lignin levels used in our analysis were obtained from wood cores, and not sampled from leaves, though the levels are typically in constant proportion among a range of plant species [40].

***Targeted Metabolomic Analysis in Low-Lignin P. deltoides***—As summarized in Table 2, the *Populus deltoides* collection obtained from the BESC and that were used in our studies included several lines that targeted a variety of genes to alter lignin levels or cellulose availability. One specific line targeted the *CAD* (cinnamyl alcohol dehydrogenase) gene, which encodes the last enzyme in the phenylpropanoid pathway involved in lignin precursor biosynthesis. This line, with an attenuated copy of the single *CAD* gene, was used to assess the impact of UV-B radiation both ecophysiologically and metabolically. Simplistically, one might expect a relatively straightforward outcome resulting from a block in this biosynthetic step: in *CAD* knockdown plants with compromised lignin, it was expected that phenylpropanoid and flavonoid levels would increase, which could be readily measured by LC-MS. The BESC used RNAi technology to attenuate the expression of the single *CAD* allele in *P. deltoides*. Inter-variance of the transgenic RNAi *CAD* lines was categorized as either a Comparator line (low transgene expression) or as a top-performing line (TOP—high transgene expression) [13]. These plants were propagated in the greenhouse at College Park and used in UV-B acclimation experiments.

Strikingly, as seen in Figure 5, UV-B treated *P. deltoides* lines downregulated in *CAD* showed remarkable anthocyanin hyper-pigmentation in newly emerging and fully expanded leaves throughout the stems, petioles, and veins as well as the leaf margin compared to the control. Motivated by this simple visual readout of UV-B acclimation, LC-MS analysis was used to determine the levels of several key relatively abundant phenylpropanoid metabolites in these plants upon exposure to UV-B radiation. Thus, quercetin glucoside, 4-coumaroyl glucoside, and several salicortin derivatives were selected for analysis since, as can be seen in Figure 6, they report on the three main branches in the biosynthesis of phenylpropanoid metabolites that may affect UV-B protection.

As can be seen in Figure 7, LC-MS analyses indicated that quercetin glucoside is an abundant flavonoid for which levels (as measured by normalized intensity) increased slightly in TOP lines relative to Comparator lines in the absence of UV-B radiation, but for which metabolite levels increased significantly in the presence of UV-B. Similarly, in Figure 8, LC-MS measurements indicated that 4-coumaroyl glucoside levels increased in both the Comparator and TOP lines relative to control lines when treated with UV-B radiation. In the absence of UV-B radiation, the Comparator lines exhibited slightly greater levels of 4-coumaroyl glucoside levels relative to TOP lines. Additionally, as shown in Figure 9A–C, salicylate levels were affected by UV-B radiation with similar trends in 2-acetylsalicortin, HCH-salicortin, and salicortin. Comparator lines showed an increase in salicylates in UV-B treated plants relative to controls. However, there were no differences seen in the TOP lines in the absence or presence of UV-B radiation.

## 4. Discussion

One challenge in developing improved, low-lignin plant feedstocks for use as biofuel or in the development of bioproducts is a full and systematic understanding of the molecular mechanism for UV-B acclimation in bioenergy crops. Here, we used ecophysiological and targeted metabolomic approaches to glean clues on the metabolic responses of *Populus* spp. upon treatment with UV-B radiation. Ecophysiological measurements were performed on natural *Populus trichocarpa* variants in an effort to correlate photo-shielding with growth elevation or with differing lignin content. Because of the considerable variation of responses to UV-B from the range of naturally occurring *Populus* variants investigated, the complex dependence of UV shielding on plant lignin levels or composition or on growth elevation suggests a range of metabolites may be synthesized, even in the absence of a stressful level of UV-B radiation, to compensate for any deficiencies in the acclimation process in plants. Alternatively, analyses of targeted phenylpropanoid metabolites in well-defined transgenic *Populus deltoides* variants suggest that the levels of protective metabolites are elevated and generally increase after UV-B treatment. The ability of these key metabolites to absorb UV-B radiation, this metabolic reprogramming may offer a protective effect and promote efficient acclimation, especially in poplars used as biofuel feedstocks that are downregulated for lignin.

Epidermal shielding measurements of leaves at LPI 2–3, LPI 4–6, and LPI 7–8 that were treated with UV-B radiation over seven days did not reveal any meaningful relationships between leaves at different growth stages and UV-B acclimation at low doses of radiation. On the other hand, it has been reported in the literature that there are some minor metabolic differences in both *Brassica* species and in *P. trichocarpa* between leaves at LPI 2–3 and LPI 7–8 when treated with UV-B radiation for long periods, including changes in the concentration of phenolics, including salicylates and flavonoids [41,42]. Younger leaves may be more sensitive to UV-B radiation as compared to older leaves because there is less accumulation of UV-absorbing compounds in vacuoles and cell walls [43,44,45]. Although these observations are interesting, the differences reported are minor, and differentiating them and their influence on the acclimation process in *Populus* will require further, more comprehensive investigation.

Natural *Populus* variants treated with higher doses of UV-B radiation (20–35 kJ m^−2^) showed an increase in epidermal UV shielding and increased photosynthetic efficiency (Fv/Fm) compared to low dose exposure (10 kJ m^−2^). Although both treatments resulted in a consistently measurable acclimation response within a few days, with a maximal effect reached in approximately 10 days, higher levels of radiation were used in this study to provide statistical support for our conclusions. The acclimation response is thought to be due in part to the accumulation of UV-B absorbing compounds in the epidermis, as well as from protection from lignin, which also arises from metabolites produced in the phenylpropanoid pathway.

Interestingly, in the population of natural variants with a low elevation origin of growth, low lignin genotypes showed a greater increase in UV shielding relative to high lignin genotypes, consistent with the rationale that low-lignin *P. trichocarpa* is more sensitive to UV-B radiation and therefore mount a more substantial metabolic response. This effect may be due to the reduced absorption of UV-B in the low lignin variants, thus necessitating the synthesis of UV-B absorbing compounds to alleviate radiation stress.

Alternatively, the high elevation origin of growth variants responded inversely to UV-B treatment. In these specimens, the low lignin plants showed a reduced level of UV shielding relative to the high lignin plants. This surprising observation suggests that UV-absorbing phenylpropanoid precursors may be abundant in low-lignin plants, and other UV-absorbing compounds are synthesized, which would have the potential to absorb greater intensities of UV light relative to lignin. The complex chemical and functional attributes of secondary metabolites in the leaf epidermis may well attenuate transmission of radiation into the mesophyll. For example, some metabolites are known to protect photosynthetic machinery or mitigate reactive oxygen species during stress [46,47]. A comprehensive, non-targeted analysis of the metabolome of well-defined *Populus* variants subjected to UV-B may help characterize the metabolic flux through the phenylpropanoid pathway that arises from radiation stress and could provide a framework to more fully understand the acclimation of low-lignin *Populus* to UV-B radiation.

Despite the complexities that underlie lignin content and composition, growth elevation, and their influence on acclimation to UV-B, it is clear that functional metabolic changes do occur in the leaves of a variety of *Populus* genotypes in response to UV-B radiation. Lignin is abundant in vascular transport structures, but it has also been detected spectroscopically in fresh leaves from various species [48]. Additionally, altered levels of monolignols can affect a spectrum of secondary metabolite precursors in the phenylpropanoid pathway [49]. Thus, in this broad context, it is possible to interpret the increases in key UV-absorbing compounds measured here, including flavonoids and salicylates, for *Populus* variants treated with UV-B. The relative increases in flavonoids and salicylates seen for Comparator and TOP lines regardless of UV-B treatment are most likely due to interactions in lignin biosynthetic components that result in metabolite alterations [49]. Furthermore, changes in metabolite flux have been demonstrated in downregulated *CAD* transgenic *Populus* that results in red pigmentation, which is attributable to increased levels of anthocyanins [50], and which is similar to the changes seen in this study. Further work is being directed to measure total salicylate levels, particularly because this class of metabolites is labile and may accumulate in leaves, stems, and shoot tips [51]. Our results suggest that metabolites in the lignin, flavonoid, and salicylate branches of the phenylpropanoid pathway are important components of the acclimation response in *Populus*. Additional genetic modification of these pathways in *Populus* with the purpose of increasing key metabolites for additional UV-B protection in low-lignin plants should provide further insight into developing new biofuel feedstocks.

## Figures and Tables

**Figure 2 metabolites-12-00767-f002:**
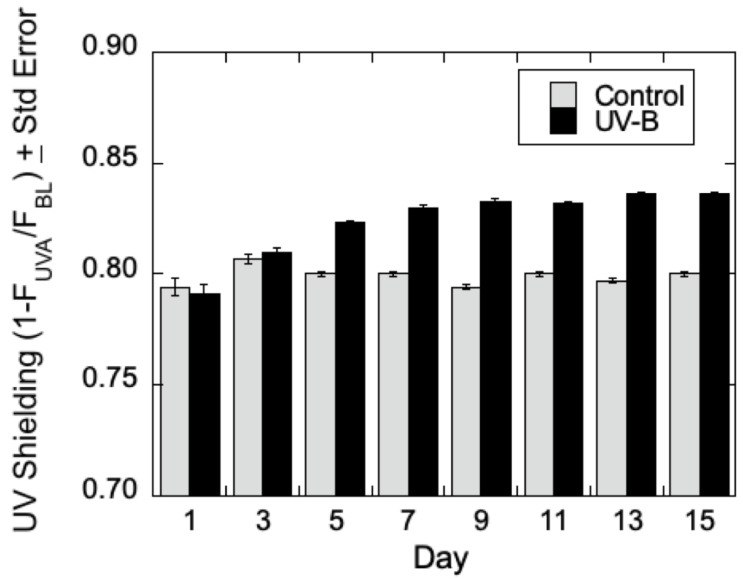
Increased UV epidermal shielding in leaves of UV-B treated *Populus deltoides.* A UV-A PAM fluorimeter was used to measure epidermal excitation radiation of chlorophyll in the mesophyll (*n* = 3) over 15 days. UV-shielding (%) = 100 (1 − F_UVA_/F_BL_) shows that UV-B treated *Populus deltoides* displayed a decrease in transmittance that is inversely proportional to increased UV protection as compared to control plants.

**Figure 3 metabolites-12-00767-f003:**
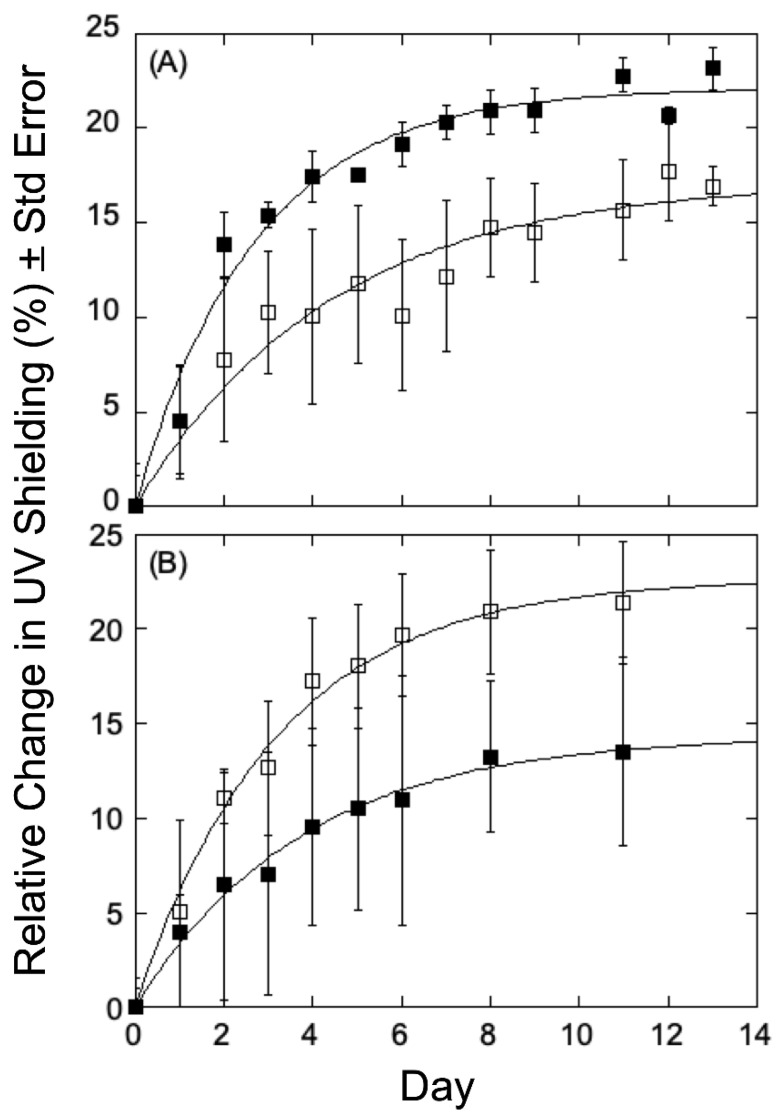
Relative change in epidermal shielding in *Populus trichocarpa* upon exposure to UV-B radiation. (**A**) Low elevation origin of growth plants showed a maximal response to UV-B (acclimation) occurs within 9–10 days of radiation exposure. The low-lignin variant (BESC-99, filled squares) showed an increased change compared to the high lignin variant (GW-10993, open squares). (**B**) High elevation origin of growth plants showed a maximal response to UV (acclimation) within 9–10 days of UV-B radiation. The high-lignin variant (GW-9889, open squares) showed an increased change compared to the low lignin variant (GW-9791, filled squares).

**Figure 4 metabolites-12-00767-f004:**
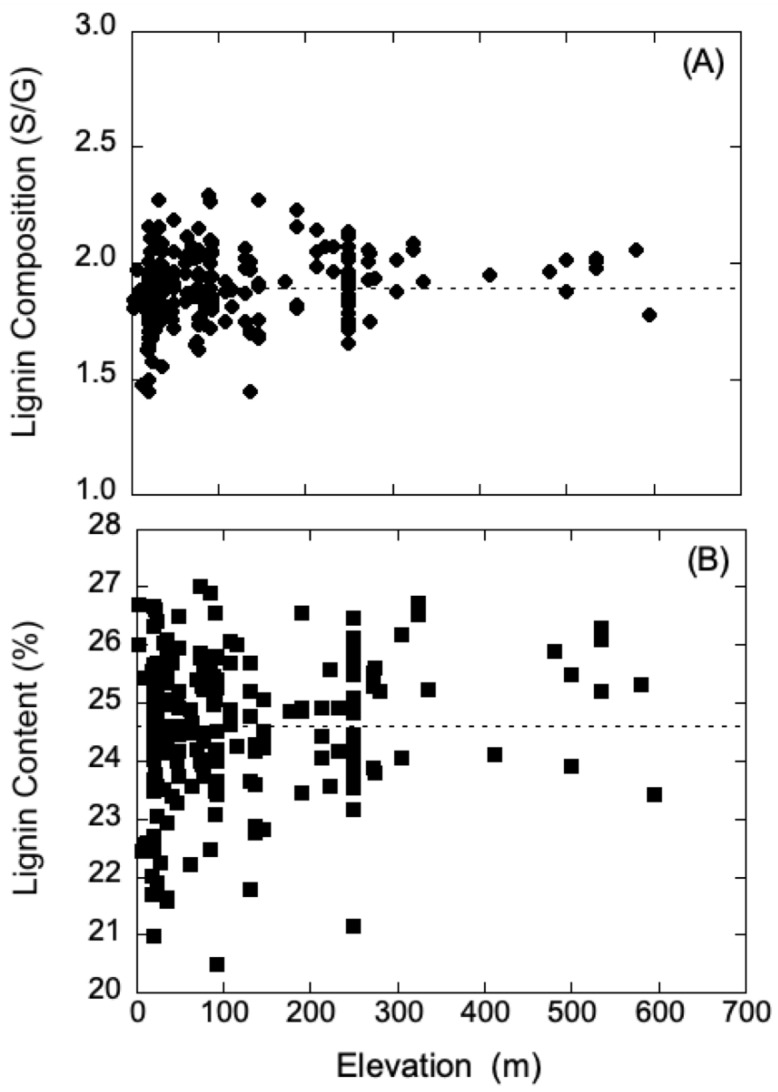
Relationship of lignin composition and content to plant origin of growth elevation. A dataset of natural variants of *Populus trichocarpa* analyzed for lignin content (%) and composition expressed as the S/G ratio [12] was used to evaluate whether geographical parameters influenced lignin attributes. (**A**) Least-squares analysis indicates that the S/G ratio shows no dependence on the origin of growth elevation. (**B**) Least-squares analysis indicates that the lignin content shows no dependence on the origin of growth elevation.

**Figure 5 metabolites-12-00767-f005:**
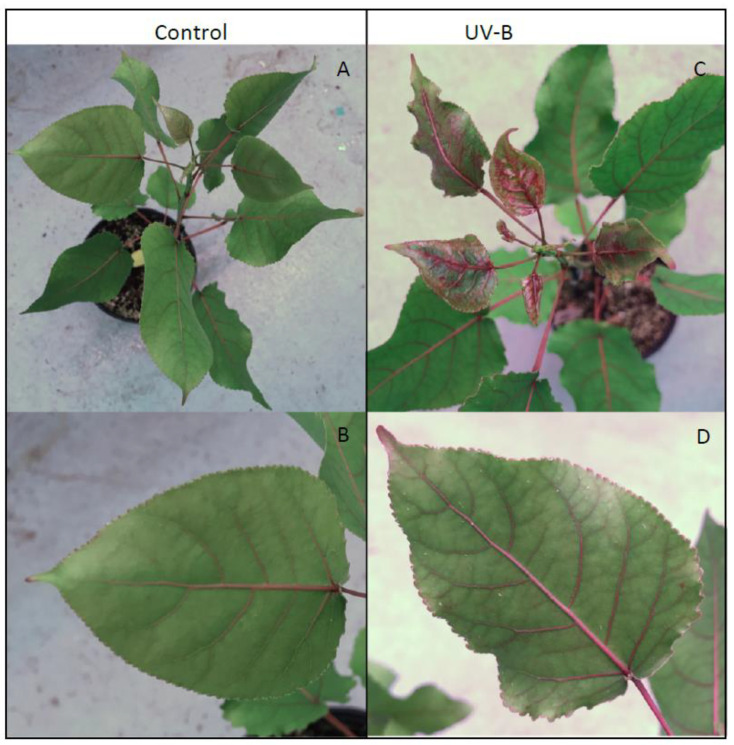
Low-lignin *Populus deltoides* show increased anthocyanin accumulation with UV-B exposure. (**A**) UV-B treatment revealed that the low-lignin *P. deltoides* control group displayed red pigmentation in the stems and petioles and (**B**) through part of the midrib and veins. (**C**) Low-lignin *P. deltoides* showed an increased accumulation of anthocyanin in the newer leaves when treated with UV-B radiation. (**D**) Older leaves show increasing anthocyanin accumulation through the veins and on the leaf margins.

**Figure 6 metabolites-12-00767-f006:**
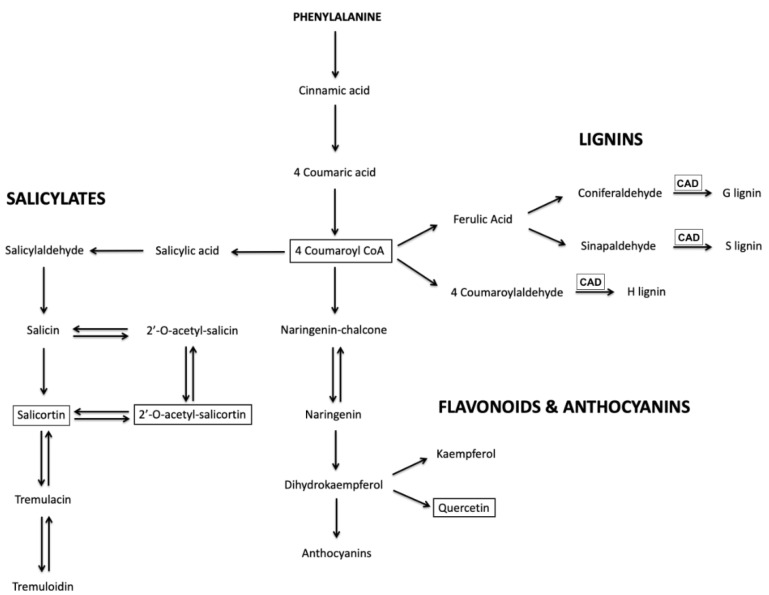
An abridged map of the phenylpropanoid pathway to highlight major branches of interest including: lignin, flavonoid, anthocyanin, and salicylate biosynthesis. Targeted metabolomic analysis was used to identify compounds in these pathways to measure changes in key metabolites after UV-B radiation treatments. Key metabolites targeted in analysis are indicated in boxes.

**Figure 7 metabolites-12-00767-f007:**
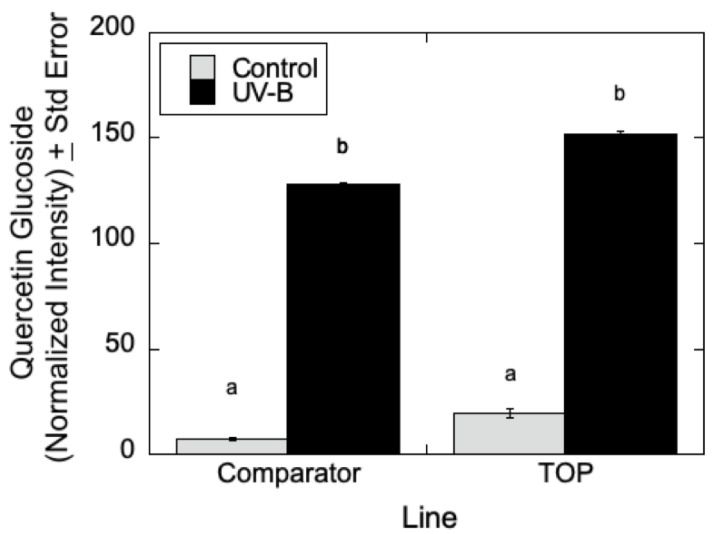
Quercetin glucoside in *P. deltoides* with differential transgene expression of *CAD*. Targeted metabolomic analysis (*n* = 3) using LC-MS show increased levels of quercetin glucoside in the Comparator (b) and TOP (b) lines treated with UV-B (*p* < 0.00) relative to Comparator and TOP controls (a), which showed no difference (*p* < 0.79 and 0.31). Statistically significant differences in normalized intensity values between Comparator and TOP lines are indicated by different letters (a and b: analysis of variance with post hoc Tukey’s HSD test, *p* < 0.05). Inferential error bars denote the standard error in the variability of the mean.

**Figure 8 metabolites-12-00767-f008:**
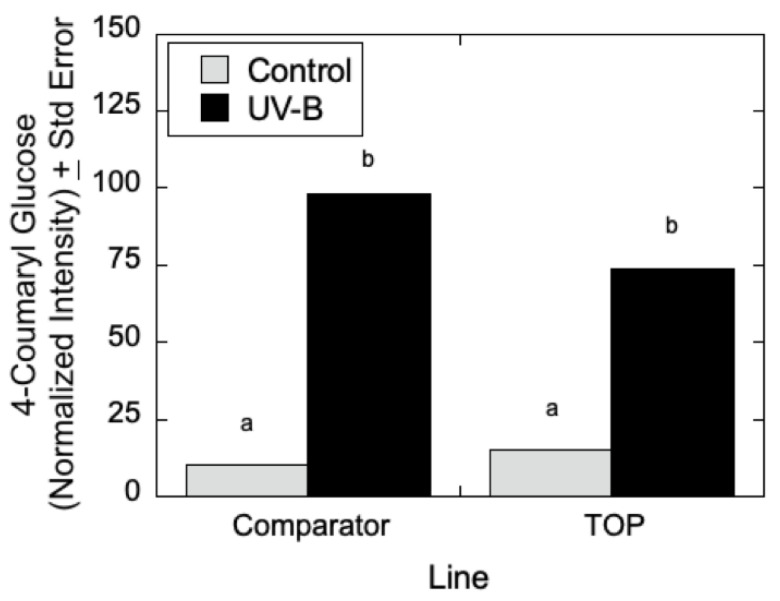
4-Coumaroyl glucose in *P. deltoides* with differential transgene expression of *CAD*. Targeted metabolomic analysis (*n* = 3) using LC-MS show increased levels of 4-coumaroyl glucose in the Comparator (b) and TOP lines treated with UV-B (b) relative to Comparator and TOP controls (a) (*p* < 0.00). Control poplars showed no difference (*p* < 0.98 and *p* < 0.25). Statistical significance in normalized intensity values between Comparator and TOP lines are indicated by different letters (a, b: analysis of variance with post hoc Tukey’s HSD test, *p* < 0.05). Inferential error bars denote the standard error in the variability of the mean.

**Figure 9 metabolites-12-00767-f009:**
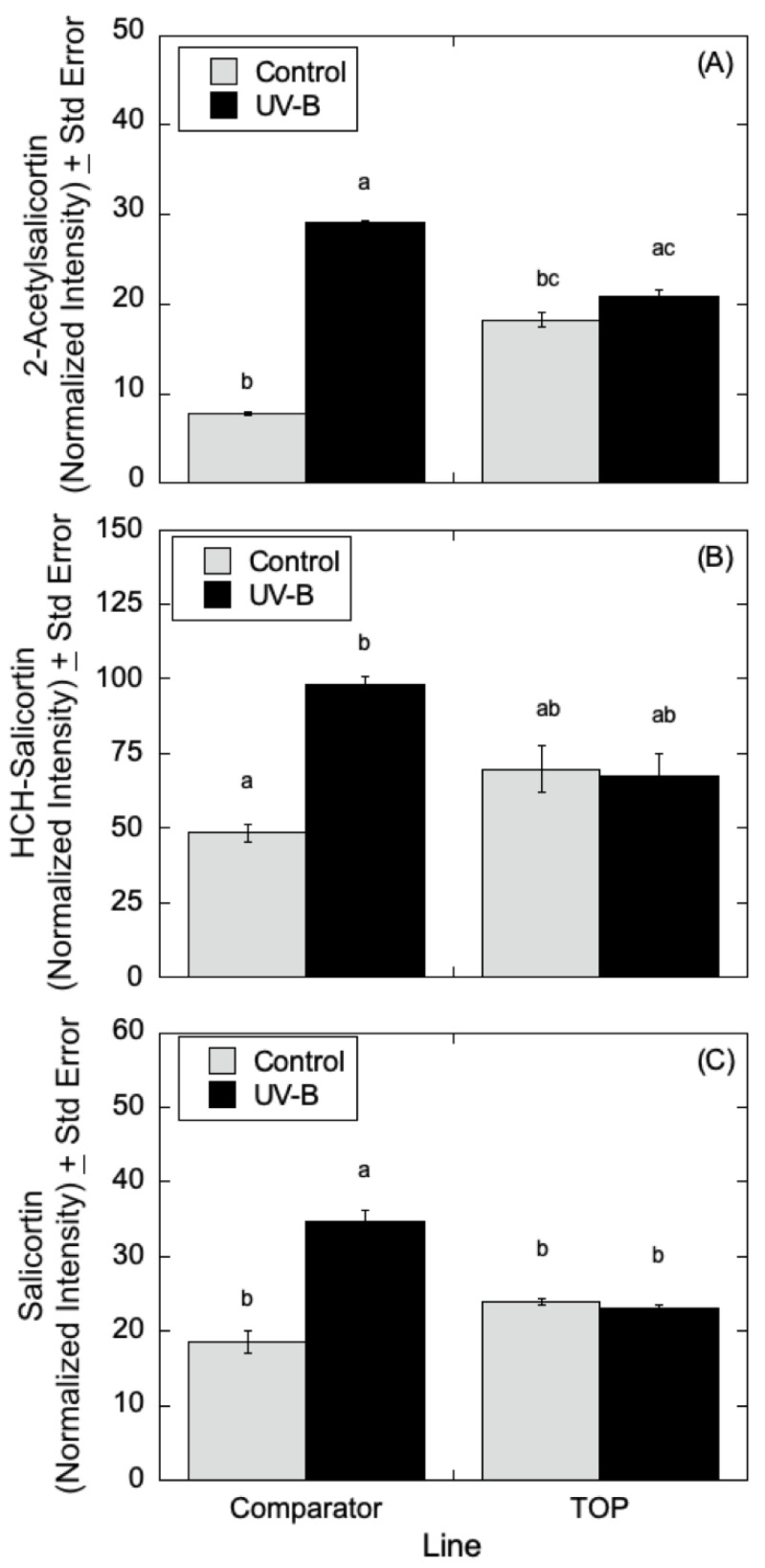
Salicylate levels in *P. deltoides* with differential transgene expression of *CAD*. (**A**) 2′-Acetylsalicortin analysis (*n* = 3) using LC-MS showed that the control Comparator line (b) was statistically different than UV-B treated Comparator (a) (*p* < 0.00), but that the control TOP (bc) was statistically identical to UV-B treated TOP (ac) (*p* < 0.91). (**B**) HCH-salicortin analysis (*n* = 3) using LC-MS showed increased levels of HCH-salicortin only in the Comparator line with UV-B treatment (b) as opposed to the control (a) (*p* < 0.01), and no difference in the TOP lines (ab) (*p* < 1.00). Additionally, no significant difference was observed between the control poplars and the UV-B treated TOP line. Statistical significance in normalized intensity values between Comparator and TOP lines are indicated by different letters (a and b: analysis of variance with post hoc Tukey’s HSD test, *p* < 0.05). Inferential error bars denote the standard error in the variability of the mean. (**C**) Salicortin analysis (*n* = 3) using LC-MS indicated increased levels of salicortin in the Comparator line with UV-B treatment (a) compared to the control (b) and TOP line (*p* < 0.04 and 0.02), but no difference in the TOP lines with or without UV-B treatment (b). For all analyses, normalized intensity values between Comparator and TOP lines are indicated as statistically significant by different letters (a, b, c and/or d: analysis of variance with subsequent Tukey’s HSD test, *p* < 0.05). Inferential error bars denote the standard error in the variability of the mean.

**Table 1 metabolites-12-00767-t001:** Characteristics of *Populus trichocarpa* genotypes with varying in lignin content and composition and origin of growth elevation.

Genotype	Characteristic	Lignin (%) ^A^	S/G ^B^	Elevation (m)
GW-10993	High lignin	27.20	2.08	11.00
GW-9911	High lignin	27.71	1.96	60.96
CHWK-27-2	High lignin	27.94	2.00	280.00
GW-9889	High lignin	27.10	1.95	365.76
HOPF-27-5	High S/G	25.32	2.31	61.00
KTMB-12-3	High S/G	25.66	2.48	61.00
GW-9827	High S/G	26.24	2.29	76.20
VNDL-27-5	Intermediate	24.45	1.93	20.00
BESC-46	Intermediate	24.18	1.71	21.30
BESC-25	Intermediate	24.44	1.95	22.40
GW-9821	Intermediate	23.31	1.77	76.2
BESC-436	Intermediate	24.11	1.80	92.57
KLNG-20-5	Intermediate	24.28	1.83	105.00
GW-9874	Intermediate	26.32	1.98	152.40
BESC-166	Intermediate	25.65	1.72	249.77
BESC-838	Intermediate	24.39	1.58	249.77
BESC-99	Low lignin	20.02	1.45	19.48
BESC-334	Low lignin	19.17	1.79	47.93
BESC-193	Low lignin	19.81	1.60	249.77
GW-9791	Low lignin	20.38	1.66	304.80
GW-10985	Low S/G	21.05	1.38	11.00
BESC-81	Low S/G	20.98	1.45	19.48
BESC-459	Low S/G	22.92	1.41	76.77
BESC-328	Low S/G	23.59	1.45	136.47
CA-01-03 ^C^	Unknown	Unknown	Unknown	2144.00

(^A^) Composite estimate of lignin composition from pyrolysis and molecular beam mass spectrometry (PyMB/MS), values were adjusted for variability within the plantation due to environment. A total of 15 genotypes were selected based on extreme values in lignin content (15.7–27.9%) and composition (1.0–3.0) in addition to 10 genotypes with average values. (^B^) Corresponds to the ratio of syringyl to guaiacyl lignin also derived from PyMB/MS. (^C^) Genotype does not have lignin chemistry data [22].

**Table 2 metabolites-12-00767-t002:** Characteristics of *Populus deltoides* genotypes affected in lignin biosynthesis via differential transgene expression.

Gene Name	Accession Number	Inter-Line Variance	Type	Annotation
PtCAD2359	Potri.009G095800	Comparator	RNAi	Cinnamyl alcohol dehydrogenase
PtCAD2359	Potri.009G095800	TOP	RNAi	Cinnamyl alcohol dehydrogenase
PtEPSPS	Potri.002G146400	Comparator	OX	5-enolpyruvylshikimate-3-phosphate synthase
PtEPSPS	Potri.002G146400	TOP	OX	5-enolpyruvylshikimate-3-phosphate synthase
PtRWA2	Potri.010G148500	Comparator	OX	*O*-acetyltransferase
PtRWA2	Potri.010G148500	TOP	OX	*O*-acetyltransferase
PtSHMT10969	Potri.001G320400	Comparator	OX	Serine hydroxymethyltransferase
PtSHMT10969	Potri.001G320400	TOP	OX	Serine hydroxymethyltransferase
PtGAUT12	Potri.001G416800	Comparator	RNAi	Galacturonosyltransferase 12
PtGAUT12	Potri.001G416800	TOP	RNAi	Galacturonosyltransferase 12
PtVND6	Potri.015G127400	Comparator	OX	Vascular-related NAC-domain 6 TF
PtVND6	Potri.015G127400	TOP	OX	Vascular-related NAC-domain 6 TF
PtDUF266	Potri.011G009500	Comparator	OX	Protein of unknown function
PtDUF266	Potri.011G009500	TOP	OX	Protein of unknown function

Identified genes involved in cell-wall development with corresponding accession numbers were targeted to alter lignin in *Populus deltoides*. A description and type of modification for each line had a Comparator and top-performing line (TOP) to account for inter-variance of the transgenic lines due to genetic instability from DNA methylation and retrotransposons induced from tissue culture plants [13].

## Data Availability

The data presented in this study are available in article.

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
