# Peer review of "Acclimation and Compensating Metabolite Responses to UV-B Radiation in Natural and Transgenic Populus spp. Defective in Lignin Biosynthesis"

_metabolites, 2022, doi:10.3390/metabo12080767_

Round 1

Reviewer 1 Report

The scientific article is valid. ACCEPTED FOR PUBLICATION

Author Response

No specific comments needed.

Reviewer 2 Report

The article entitled "Acclimation and compensating metabolite responses to UV-B radiation in natural and transgenic Populus spp. defective in lignin biosynthesis" reports the ecophysiological and metabolite profiling approaches of the responses of Populus spp. upon treatment with UV-B radiation.

This article contains too much confusion and approximation to be accepted. Under these conditions it is difficult to understand the scientific approach.

Major comment:

-          the experimental design is not clear. A large number of Populus lines are available but only a part of these lines have been studied. For example, the Figure 2 represents the epidermal shielding in leaves of UV-B treated Populus deltoides over a 15-day interval. It is not indicated which lines have been studied. It is also not clear why the results for Populus trichocarpa were not shown. These confusions in the samples analysed are also observed in the figure 3, 4, 7, 8 and 9. Without these precisions it is difficult to understand the scientific approach.

minor comments:

-          Figure captions contain too much detail. Some information should be placed in the material and method part or in the results part.

-          The choice of metabolites whose content has been monitored must be better argued.

-          change “metabolomic” by “targeted metabolomics”.

-          via must be in italics in all the document.

-          Populus must be in italics for keywords and page 9.

-          change “p-coumaryl alcohol” by “p-coumaryl alcohol” (page2).

-          change “O-acetyltransferase” by “O-acetyltransferase” (Table 2).

-          change “UV UV-B” by “UV-B” (page 6).

-          m/z must be in italics (page 7).

-          change “LCMS” by “LC-MS” (page 7).

-          figure 2: Figure captions must be completed for Day

-          figure 3: indicate that the y-axis has been cut

-          figure 4: align figure A and B. Indicate the names of the samples.

-          for the sentence “As can be seen in Figure 4 (A), an analysis of the composition of lignin (syringyl and guaiacyl ratio) showed no relationship to growth elevation.”, what are the statistical arguments that demonstrate this lack of relationship

Reviewer 3 Report

Dear Authors,

My comments on the paper are:

- the word lignin appears twice in the key words;

- under each figure there is too much text describing it, the explanations of the figures should be in the text;

- the headings of Table 1 and Table 2 are repeated twice, above and below the tables.

Author Response

1. The word lignin appears twice in the key words;

A1. The second occurrence of lignin has been removed from the keywords.

2. Under each figure there is too much text describing it, the explanations of the figures should be in the text;

A2. The captions of all the figures have been streamlined.  However, we have maintained a lengthy legend to Figure 1 since we believe that many readers of Metabolites will be unfamiliar with ecophysiological approaches used to glean metabolic changes in plants. 

3. The headings of Table 1 and Table 2 are repeated twice, above and below the tables.

A3. The duplications in the tables have been removed.

Round 2

Reviewer 2 Report

The article has been clarified and can be accepted in the present form.